# HAMP as a Potential Diagnostic, PD-(L)1 Immunotherapy Sensitivity and Prognostic Biomarker in Hepatocellular Carcinoma

**DOI:** 10.3390/biom13020360

**Published:** 2023-02-14

**Authors:** Guoming Chen, Cheng Zhang, Danyun Li, Dongqiang Luo, Hui Liao, Peizhen Huang, Ning Wang, Yibin Feng

**Affiliations:** 1School of Chinese Medicine, Li Ka Shing Faculty of Medicine, The University of Hong Kong, Hong Kong 999077, China; 2The First Clinical School, Guangzhou University of Chinese Medicine, Guangzhou 510405, China; 3The Second Clinical School, Guangzhou University of Chinese Medicine, Guangzhou 510405, China

**Keywords:** hepatocellular carcinoma, HAMP, PD-1, immunotherapy, prognosis

## Abstract

Hepatocellular carcinoma (HCC) remains a global medical problem. Programmed cell death protein 1 (PD-1) is a powerful weapon against many cancers, but it is not sensitive to some patients with HCC. We obtained datasets from the Gene Expression Omnibus (GEO) database on HCC patients and PD-1 immunotherapy to select seven intersecting DEGs. Through Lasso regression, two intersecting genes were acquired as predictors of HCC and PD-1 treatment prognosis, including HAMP and FOS. Logistic regression was performed to build a prediction model. HAMP had a better ability to diagnose HCC and predict PD1 treatment sensitivity. Further, we adapted the support vector machine (SVM) technique using HAMP to predict triple-classified outcomes after PD1 treatment in HCC patients, which had an excellent classification ability. We also performed external validation using TCGA data, which showed that HAMP was elevated in the early stage of HCC. HAMP was positively correlated with the infiltration of 18 major immune cells and the expression of 2 important immune checkpoints, PDCD1 and CTLA4. We discovered a biomarker that can be used for the early diagnosis, prognosis and PD1 immunotherapy efficacy prediction of HCC for the first time and developed a diagnostic model, prognostic model and prediction model of PD1 treatment sensitivity and treatment outcome for HCC patients accordingly.

## 1. Introduction

Liver cancer, as one of the four main causes of cancer-related death all over the world, has brought heavy human and economic burdens to society [1,2]. Hepatocellular carcinoma (HCC) takes up the vast majority of primary liver cancer cases, which is a primary malignant neoplasm of epithelial liver cells [3]. According to a recent study using the SEER registry, the incidence of HCC is predicted to continue climbing until 2030 [4]. In the face of the low survival rate and high mortality rate of liver cancer worldwide, how to effectively reduce the burden associated with liver cancer remains a major issue in the field of global public health as well as chronic disease prevention and control, while the optimization of screening strategies for the liver cancer population is an essential goal that deserves continuous exploration.

Although the diagnosis and treatment techniques of liver cancer have developed in recent years, early detection and monitoring are still obscure [4,5,6]. Currently, alpha-fetoprotein (AFP), accompanied by ultrasound scanning (USS), multiphasic computed tomography (CT) and magnetic resonance imaging (MRI), is widely used for the surveillance and diagnosis of HCC [7]. Some tools from other serum biomarkers and liquid biopsy are also in development but require fuller clinical evaluation [8,9]. It is worth pointing out that there are no specific markers of early liver cancer and clinical stages in the progression from precancerous lesions to liver cancer [4,10,11,12].

As to therapies, while immunotherapy has played an extremely important role in the treatment of many tumors in recent years, clinical studies of immune checkpoint inhibitors (ICIs) in liver cancer have continued to emerge [13,14]. PD-1 therapy, as one of the systemic therapeutic schedules, demonstrated extraordinary success in various cancers [15]. In the background of immunotherapy gradually becoming one of the four pillars of tumor therapy, PD-1 and its immunotherapy ligands PD-L1 and PD-L2 have come into sight, which shows an encouraging effect in treating some cancers such as melanoma, non-small-cell lung cancer, renal cell carcinoma, Hodgkin’s lymphoma, bladder cancer and so on [4,15,16]. The PD-1/PD-L1 pathway is one of the mechanisms of immune escape [16]. By inhibiting this pathway, T cells can be activated to clear the tumor [16]. Nevertheless, clinical practice shows that immune checkpoint blockers such as nivolumab and pembrolizumab are not as efficacious as once expected, which may be due to the upregulated Wnt/β-catenin pathway and β-catenin pathway and a lack of intratumoral T cells [17,18,19]. Generally, the abnormal expression of key genes precedes clinicopathological abnormalities [20]. Gene expressions differ between patients who are sensitive to PD-1 treatment and those who are not, and it affects the efficacy of PD-1 among the appropriate group as well [20]. Consequently, for successful treatment progression, there is an urgent need to identify biomarkers that can steer disease monitoring and treatment decisions in early HCC patients receiving immunotherapy [21].

With the rapid development of genomics, proteomics, and metabolomics, as well as the discovery and application of PCR amplification and sequencing technology, liquid chromatography-mass spectrometry metabolomic analysis technology, and quantitative proteomics technology, the search for more accurate and effective markers in the early screening of liver cancer has become a high point of research. This study tries to identify novel biomarkers for the early diagnosis, prognosis and prediction of PD-1 immunotherapy sensitivity and development with respect to new medicines. With detection or monitoring, not only can the biomarkers facilitate early detection, but also screen suitable HCC patients and predict their prognosis [22]. Furthermore, the discovery of novel biomarkers is beneficial to developing new drugs, as the combination of regulating the expression of key genes and PD-1/PDL-1 may be promising for future liver cancer therapies [23].

## 2. Materials and Methods

### 2.1. Acquisition of Datasets

With the search criteria of “PD-1 Receptor” and “Liver Neoplasm”, the dataset GSE140901 was obtained from the Gene Expression Omnibus (GEO) of NCBI (https://www.ncbinlm.nih.gov/geo/, accessed on 2 February 2022), which is based on the GPL19965 platform and contains a total of 24 samples, all of which were treated with PD-1 immunotherapy, and they were classified as 6 with partial remission (PR), 10 with stable disease (SD) and 8 with progressive disease (PD) according to the treatment efficacy. The raw data on the “Liver Tumor” criterion were downloaded from GSE14520 based on the GPL571 platform, and 43 samples were availa2015013052@stu.gzucm.edu.cnble, of which 22 were tumor samples and 21 were normal samples. The patients back ground data was showed in Table 1, Table 2 and Table 3.

### 2.2. Extraction of Differentially Expressed Genes and Lasso Regression

The GSE14520 dataset was grouped in accordance with whether the patients had liver cancer, and the GSE140901 dataset was grouped in accordance with whether the liver cancer patients were in remission after PD-1 immunotherapy; these two datasets were analyzed for discrepancies using the DESeq2 and ggrepel software packages, and they were visualized in volcano plots at |log2FC| ≥ 1 with a *p*-value < 0.05 to screen for DEGs. Subsequently, the intersection sets of the above two differential gene sets were excavated as predictor variables and further screened via Lasso regression. Lasso regression is characterized by the ability to perform variable selection and complexity regularization while fitting a generalized linear model. Then, UpSet plots were built for the aforementioned genes so as to acquire the pivotal genes that were simultaneously associated with liver cancer disease and the prognosis of PD-1 immunotherapy in liver cancer patients. In the field of bioinformatics research, when the number of sample groups is relatively large, it is more intuitive to employ UpSet plots to observe the information that is common among all sample groups as well as the information unique to each sample group.

### 2.3. Model Construction and Validation, Results Evaluation

The two pivotal intersecting genes screened in 2.2 were constructed as logistic regression models for the GSE14520 and GSE140901 datasets, respectively. Thereafter, subject working characteristic curves were plotted using the pROC package, and the area under the curve (AUC) was calculated to assess the discriminatory ability of the prediction models. It should be highlighted that the nomograms of the critical factors were created separately for visualization in this study. Nomograms are able to visualize the results of regression analysis and allow for a more intuitive presentation of multiple indicators that combine to diagnose or predict disease risk or prognosis. Meanwhile, this study also plotted survival curves using the Kaplan–Meier (KM) method by using individual overall survival times (OS) as an outcome variable for the GSE14520 and GSE140901 datasets, separately, and performed a logrank test to compare significant differences in survival curves, aiming to screen genes that specifically affect the prognosis of PD-1 treatment.

### 2.4. Construction of Support Vector Machine Model

The support vector machine (SVM) algorithm is capable of mapping the categorized data to a higher dimensional feature space with certain fault tolerance conditions based on an appropriate kernel function and classifying the data by constructing an optimal classification hyperplane in this space. This algorithm is suitable for small sample, nonlinear, high-dimensional data classification problems, with high reliability in its prediction, stability and generalization ability, and it is widely applied to processing classification problems and regression analysis. However, the model built in this study incorporates only one feature variable and does not require an SVM algorithm for high-level processing data; compared with logistic regression, SVM is more suitable for small-scale data sets. Moreover, SVM is better for handling nonlinear relationships; therefore, it is feasible for us to model with the SVM algorithm. The present study segmented the GSE140901 dataset into three categorical outcome variables on the basis of post-treatment efficacy, constructing prediction models of pivotal genes screened from 1.3 by SVM and assessing them through the subject working characteristic curves.

### 2.5. Validation of Pivotal Gene in TCGA Database

To provide a further test of the actual value of the pivotal gene in clinical samples, we downloaded liver cancer and its normal samples from the TCGA database (https://www.cancer.gov/aboutnci/organization/ccg/research/structural-genomics/tcga, accessed on 2 February 2022). The patients’ background for TCGA can be found in Table 4. Furthermore, as the prognosis of HCC is determined by liver function, we also provided the Child–Pugh scores of these patients in the background data. Benefiting from the R software (version 4.0.2, https://www.Rproject.org, accessed on 2 February 2022) and its powerful data analysis capabilities and box plotting in the “ggplot2” R package, we compared the expression levels of the pivotal gene in normal versus HCC tissues, as well as in different tumor stages. The expression differences between HCC and normal tissue samples were identified with a *p*-value of <0.05.

Sequentially, the pROC package was employed to perform sensitivity- and specificity-based ROC curve analyses, which were designed to assess the accuracy of the pivotal gene diagnosis. Overall survival (OS), disease-specific survival (DSS) and progression-free interval (PFI) were deemed to be indicators to explore the correlation between pivotal gene expression and HCC patient prognosis. When it concerned survival analysis, the Kaplan–Meier method and logrank test were deployed.

What is more, the correlations between the pivotal gene and the 24 immune cells’ infiltration levels in HCC were assessed using CIBERSORT (https://cibersort.stanford.edu/, accessed on 2 February 2022) and the R software packages “ggplot2” and “ggpubr”. As a final step, we separately evaluated the co-expression of the pivotal gene with PDCD1 and CTLA4 in HCC using Spearman correlation coefficients, which are the target genes of two well-recognized anticancer miracle drugs.

## 3. Results

### 3.1. Extraction of Differential Genes and Lasso Regression

Through differential analysis, 1154 DEGs were screened in GSE14520, of which 635 were upregulated and 519 were downregulated (Figure 1A); 109 DEGs were screened in GSE140901, of which 56 were upregulated and 53 were downregulated (Figure 1B). The intersection of the aforesaid two differential genes was taken to obtain seven genes, which were considered predictor variables for Lasso regression; thus, the λ value with the smallest cross-validation error was selected via ten-fold cross-validation, resulting in finding the best predictor by continuously fitting the model. Under Lasso regression (Figure 1C,D), we shortlisted four predictors for HCC including FOS, HAMP, IL13RA2 and LCN2, as well as four predictors for the prognosis of the PD-1 treatment, including SPP1, FOS, CXCL2 and HAMP, ulteriorly cross-tabulating the two to obtain two genes, HAMP and FOS (Figure 1E).

### 3.2. Model Building and Validation, Evaluation of Results

The two pivotal genes identified in Section 3.1 were regarded as independent variables for logistic regression to build a prediction model. In this study, the prediction performance was evaluated using ROC curves, and nomograms were drawn using RMS. As a general rule, we use the AUC value for the evaluation of the predictive model, which corresponds to a larger AUC for better prediction. When AUC = 1, the model is considered as a perfect predictive model; when AUC = [0.85,0.95], it is regarded as a very good predictive model; when AUC = [0.7,0.85], it is deemed as an average predictive model. In GSE14520, the ROC curve (HAMP: 0.651 [0.000,0.667], AUC = 0.996; FOS: 0.618 [0.048,0.667], AUC = 0.989) was modeled as a dichotomous outcome variable with good discrimination using nomograms (Figure 2A–D); in GSE140901, the ROC curve was modeled as a dichotomous outcome variable with the ability to remit after PD-1 immunotherapy (HAMP: 0.446 [0.056,0667], AUC = 0.824; FOS: 0.261 [0.222,0.667], AUC = 0.75) with good discrimination (Figure 3A–D). Additionally, after KM curve drawing and logrank analysis (Figure 3E,F), it was surprising to discover that HAMP had a better prognostic performance in PD-1 treatment (*p* < 0.05).

### 3.3. Construction of Support Vector Machine Model

Setting HAMP as an independent variable, we constructed an SVM model utilizing the e1071 package and applied a lattice parameter search function to find the best cost value. When cost = four, we constructed the best support vector machine model, and the performance of the model was assessed using multi-category ROC curves (PD: 2.50 [0.20,1.00], AUC = 0.9; PR: 2.50 [0.00,0.20], AUC = 0.6; SD: 2.50 [0.00,1.00], AUC = 1.0). Meanwhile, the model also had good discriminative performance in the prognostic multiclassification (Figure 4).

### 3.4. The Clinical Value of the Pivotal Gene in the TCGA Database

Based on the TCGA data, by comparing the expression levels of HAMP in normal samples with those of HCC samples, it was obvious that the expression level of HAMP was significantly reduced in HCC tissues (Figure 5A). This conclusion was also applicable in the comparison of paired samples (Figure 5B). When examining the relevance of the tumor stage, we noticed that the expression of HAMP displayed a striking decrease in the early stage of HCC (Figure 5C–E). When focusing on Figure 6, the AUC was up to 0.946, strongly demonstrating the high accuracy of the HAMP prediction model in the diagnosis of HCC.

The survival association analysis for HCC illustrated that the expression levels of HAMP were associated with DSS (*p* < 0.05) (Figure 7B). DSS is a good response to the clinical benefit of a specific disease, that is, the reduction or increase in death due to a specific disease. Hence, we configured a prognostic prediction model for DSS to assess the survival probability at 1, 3, and 5 years for patients with HCC, which suggested that a higher HAMP level tends to be associated with a better prognosis (Figure 7D).

From the results of CIBERSORT, it appeared that for HCC, the level of immune cell infiltration was remarkably correlated with HAMP expression to a great extent, except for Tcm (Figure 8). Meanwhile, the infiltration level of B cells, macrophages, and iDC was most closely correlated with HAMP expression (Figure 8). Furthermore, the results in Figure 9 leave no doubt about the significant co-expression relationship that HAMP has with PDCD1 (*p* < 0.001) and CTLA4 (*p* = 0.004). Immunotherapy targets mainly included CD274, PDCD1, and CTLA4. CD274 correlation was calculated as cor = 0.066, *p* = 0.203 > 0.05, so it was not included. Moreover, CD8, T-cell failure, tumor-associated fibroblasts (CAF), and tumor-associated macrophages (TAM) markers belonged to tumor microenvironment markers and were not in our scope of concern.

## 4. Discussion

In the present study, firstly, we attained two crossover genes, HAMP and FOS, by differential gene extraction and lasso regression analysis. Next, to pick a more valuable biomarker, we employed logistic regression to build prediction models using HAMP and FOS as independent variables. Combining the results of ROC curve and nomogram, we concluded that HAMP has a better ability to diagnose HCC and predict PD-1 treatment sensitivity. As a further step, we constructed the SVM model of HAMP to predict the three-stage outcome after PD-1 treatment in HCC patients, which had good classification ability. In addition, we performed external validation by using TCGA data, which revealed that the expression level of HAMP was significantly reduced in HCC tissues and was significantly elevated in the early stage of HCC, suggesting an important role of HAMP in the early diagnosis of HCC. More notably, we found that HAMP was significantly and positively correlated with the expression of 18 major immune cell infiltrates and 2 important immune checkpoints, PDCD1 and CTLA4. This finding lays a foundation for HAMP to guide the immunotherapy of HCC, which could be beneficial for clinical precision treatment and saving medical resources.

Hepcidin antimicrobial peptide (HAMP), a liver-produced protein-coding gene, is the master regulator of iron homeostasis maintenance [24], and it can be detected in human urine and serum [25]. Under normal physiological conditions, HAMP can function by promoting endocytosis and the degradation of ferroportin/SLC40A1, resulting in the retention of iron in iron export cells and reduced iron influx in plasma [26,27,28]. In previous studies, HAMP has been regularly associated with hemochromatosis, especially type 2B hemochromatosis [29]. This is significantly attributed to the metabolism of iron in the placenta and the effect of Hfe on hemoglobin production [30,31]. Likewise, in this research, we found that HAMP also has a non-negligible role in the diagnosis of HCC based on analysis of variance, Lasso regression and model construction. In Figure 2A,B, the areas under the curve for HAMP and FOS are both over 0.98, which is higher than the performance of alpha-fetoprotein (AFP). Considering the result may be affected by the small number of GEO samples we included, as well as sample bias existing in each dataset, we used another dataset as external validation to ensure the robustness of the biomarkers. Although some bias or error in our results is inevitable due to the small sample size or the source bias present in the dataset population, we identified a biomarker with excellent sensitivity and specificity, whether reliable or not, but worthy of further clinical and basic experimental confirmation and excavation. The conclusion coincides with the notions of some other researchers; in other words, it can be corroborated with them. The work of Silvia Udali et al. highlighted the emergent role of the downregulation of HAMP induced by DNA promoter hypermethylation [32]. Other investigators have elucidated the potential of HAMP as a diagnostic biomarker for HCC from the perspectives of HAMP-SLC40A1 signaling, cyclin4-dependent kinase-1/STAT3 pathway and Circ_0004913 [33,34,35]. Since the existing studies have focused more on the mechanism of the effect of HAMP on HCC, this research was more concerned with its practical application in the clinical diagnosis and treatment process; thus, our validation analysis of the TCGA data corroborated this very favorably. Although a small proportion of the paired samples had results contrary to our findings, it is the consensus among the medical community that HCC is a multifactorial predisposing disease, so these deviations are permitted to exist.

Of more interest, however, is that this study further examines the predictive role of HAMP in assessing the sensitivity and prognosis of patients with HCC to PD-1 therapy, which is of great significance both in terms of improving effective clinical treatment rates as well as saving healthcare resources. PD-1 immunotherapy serves to increase the antitumor activity of T cells in the body by relieving the suppression of immune function, which is mediated as a result of the combination of PD-1 and PDL-1, thus providing a killing and suppressing impact on cancer cells [36]. In some cancers, for example, cholangiocarcinoma, renal cell carcinoma and colorectal cancer, the relevant evidence implies that HAMP may contribute to immune activation in the tumor microenvironment and inhibit cancer cell development by acting synergistically with immune cells and chemokines [37,38]. This may corroborate the predictive role of HAMP in liver cancer immunotherapy. Additionally, a protein dynamics analysis of iron and iron interactions during T cell activation uncovered that pathways rich in iron-interacting proteins were likely impaired by iron deficiency during T cell activation and emphasized the role of iron in T cell differentiation [39]. At this point, we reasonably conclude that the iron metabolism mechanism is an intermediate pivot linking HAMP and PD-1; nevertheless, more experimental evidence still needs to be performed. 

The main inadequacy of this study was that the superior prognosis of HAMP was not as well captured in the OS and PFI of patients with HCC. Regarding this point, we speculated that it was due to a high number of missing follow-ups in the extracted patient data; thus, the time of death of these patients was not precisely estimated. In addition, the cause of HCC is still not understood, due to the limitation of data sources.

In a nutshell, this investigation suggested whether HAMP could be considered a new biomarker for diagnosing early-stage HCC and used to predict the sensitivity and prognosis of PD-1 immunotherapy, in addition to orienting the development of new drugs.

## Figures and Tables

**Figure 1 biomolecules-13-00360-f001:**
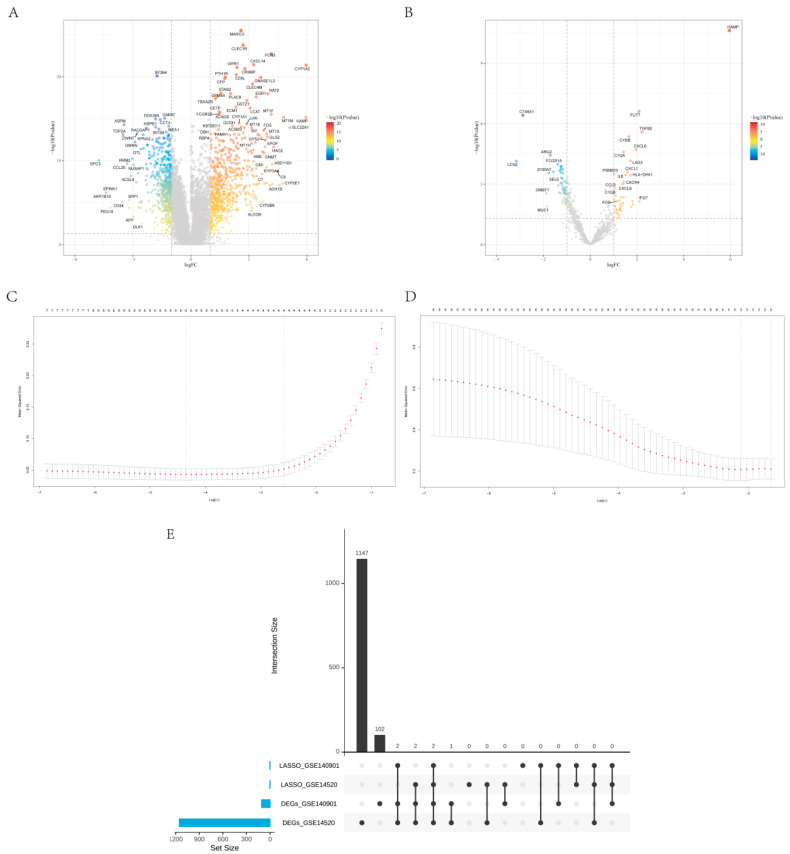
(**A**) Volcano plot of DEGs in GSE14520; (**B**) volcano plot of DEGs in GSE140901; (**C**) Lasso regression of DEGs in GSE14520; (**D**) Lasso regression of DEGs in GSE140901; (**E**) DEGs upset diagram.

**Figure 2 biomolecules-13-00360-f002:**
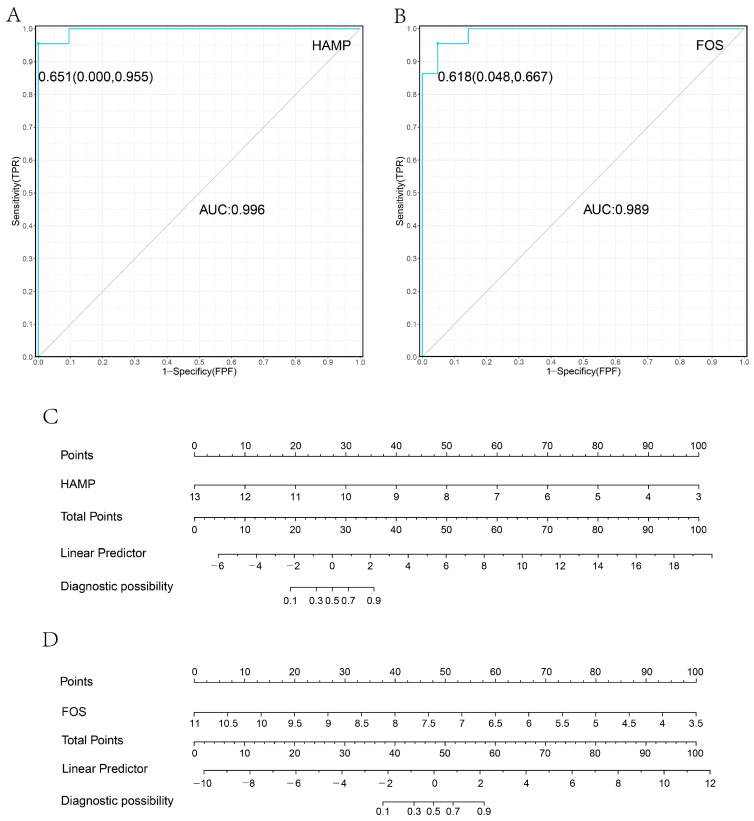
(**A**) ROC curve of HCC diagnosis model of HAMP; (**B**) ROC curve of HCC diagnosis model of FOS; (**C**) nomogram of HCC diagnosis model of HAMP; (**D**) nomogram of HCC diagnosis model of FOS.

**Figure 3 biomolecules-13-00360-f003:**
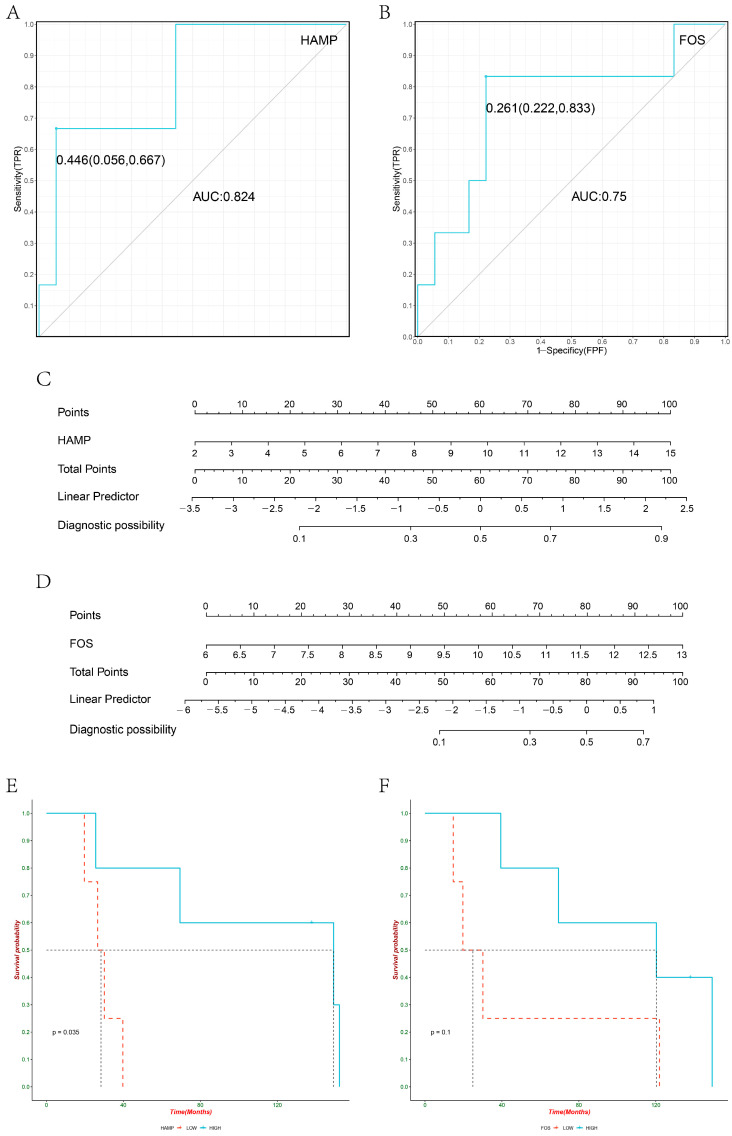
(**A**) ROC curve of HAMP predicting remission in HCC patients treated with PD-1; (**B**) ROC curve of FOS predicting remission in HCC patients treated with PD-1; (**C**) nomogram of HAMP diagnosis in HCC patients treated with PD-1; (**D**) nomogram of FOS diagnosis in HCC patients treated with PD-1; (**E**) KM curve of prognostic HAMP in HCC patients treated with PD-1; (**F**) KM curve of prognostic FOS in HCC patients treated by PD-1.

**Figure 4 biomolecules-13-00360-f004:**
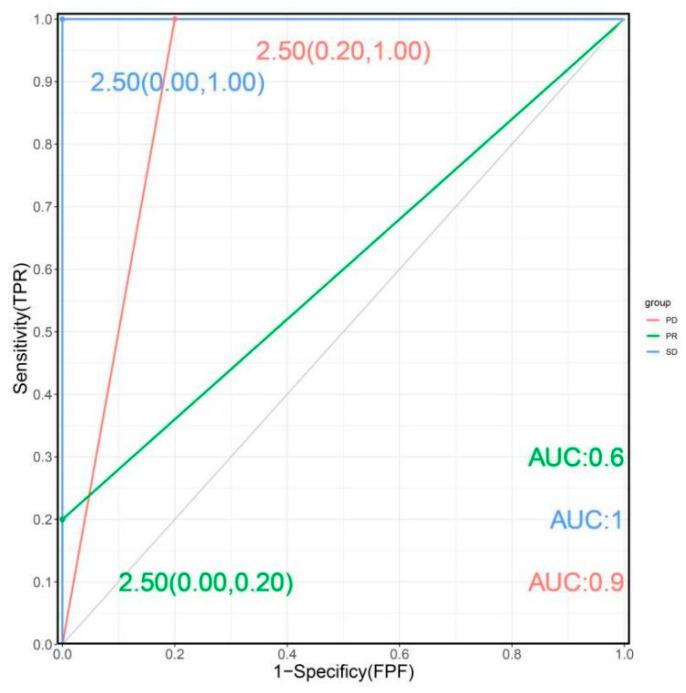
ROC curve of HAMP predicting trichotomous outcomes in HCC patients treated with PD-1 (green: PR; blue: SD; red: PD).

**Figure 5 biomolecules-13-00360-f005:**
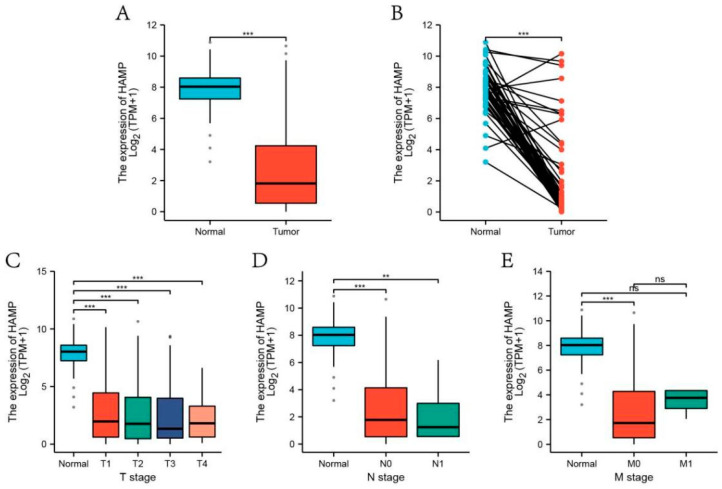
(**A**) Comparison of HAMP expression between HCC and normal samples. (**B**) Comparison of HAMP expression between HCC and paired normal samples. (**C**–**E**) Association between HAMP expression and HCC tumor stage. * *p* < 0.05, ** *p* < 0.01, *** *p* < 0.001.

**Figure 6 biomolecules-13-00360-f006:**
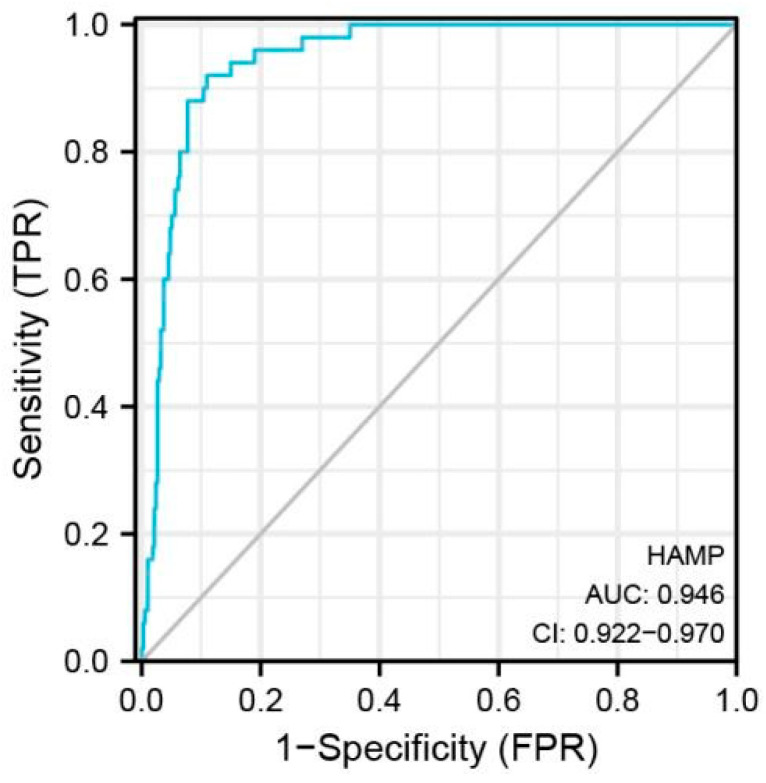
The AUC of ROC curves verified the diagnosis performance of HAMP in the TCGA.

**Figure 7 biomolecules-13-00360-f007:**
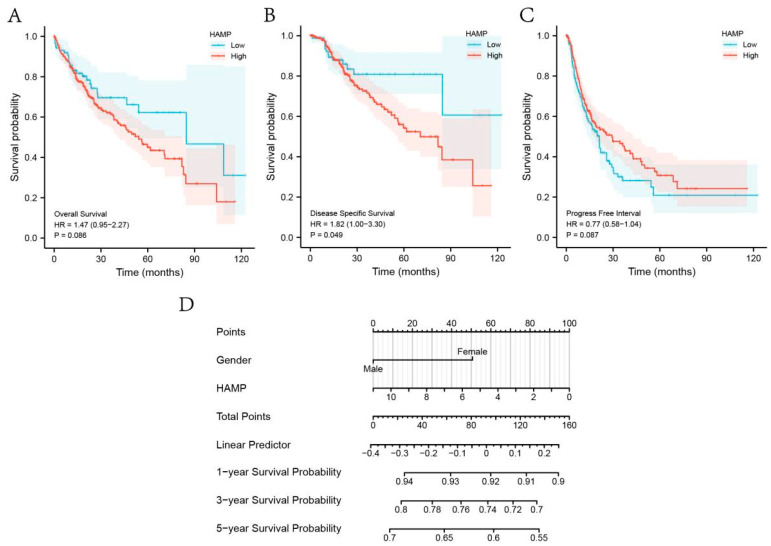
(**A**–**C**) Association between HAMP expression and OS, DSS, and PFI. (**D**) Nomogram of HCC prognosis of DSS at 1, 3, and 5 years.

**Figure 8 biomolecules-13-00360-f008:**
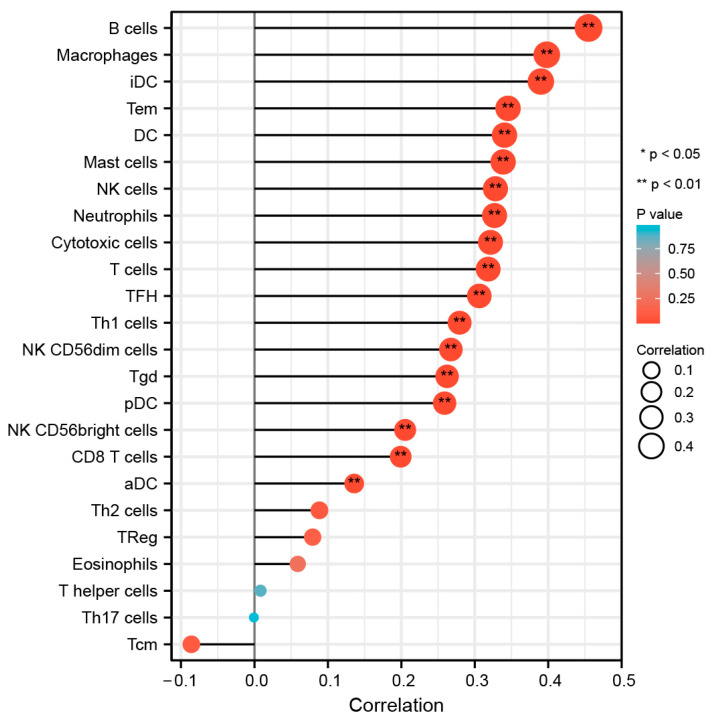
Correlation of HAMP expression with the infiltration of 24 immune cells.

**Figure 9 biomolecules-13-00360-f009:**
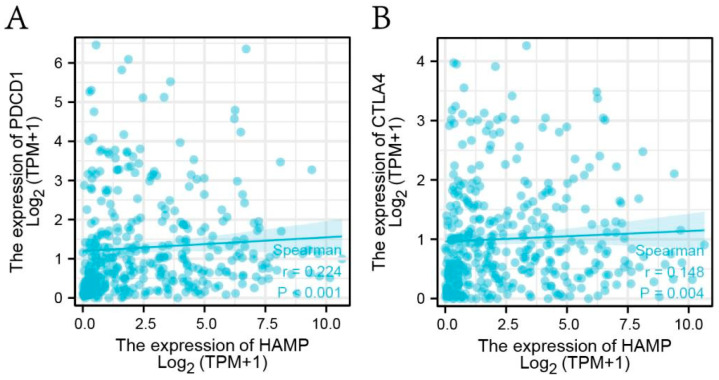
(**A**) Co-expression of HAMP and PDCD1. (**B**) Co-expression of HAMP and CTLA4.

**Table 1 biomolecules-13-00360-t001:** Patients overall back ground for GSE140901.

	Level	Overall
n		24
etiology (%)	HBV	19 (79.2)
	HCV	3 (12.5)
	Other	2 (8.3)
gender (%)	F	2 (8.3)
	M	22 (91.7)
os_event (%)	Alive	4 (16.7)
	Dead	20 (83.3)
pfs_event (%)	No progress	3 (12.5)
	Progress	21 (87.5)
os_time (mean (SD))		74.3 (51.3)
pfs_time (mean (SD))		29.6 (38.6)

**Table 2 biomolecules-13-00360-t002:** Patients PD, PR and SD back ground for GSE140901.

	level	PD	PR	SD	*p*
n		8	6	10	
etiology (%)	HBV	6 (75.0)	4 (66.7)	9 (90.0)	0.365
	HCV	1 (12.5)	2 (33.3)	0 (0.0)	
	Other	1 (12.5)	0 (0.0)	1 (10.0)	
gender (%)	F	2 (25.0)	0 (0.0)	0 (0.0)	0.113
	M	6 (75.0)	6 (100.0)	10 (100.0)	
os_event (%)	Alive	0 (0.0)	2 (33.3)	2 (20.0)	0.237
	Dead	8 (100.0)	4 (66.7)	8 (80.0)	
pfs_event (%)	No progress	0 (0.0)	2 (33.3)	1 (10.0)	0.167
	Progress	8 (100.0)	4 (66.7)	9 (90.0)	
os_time (mean (SD))		33.4 (22.6)	120.2 (47.6)	79.5 (46.3)	0.003
pfs_time (mean (SD))		5.8 (0.9)	65.9 (56.8)	27.0 (25.1)	0.009

**Table 3 biomolecules-13-00360-t003:** Patients back ground for GSE14520.

	Level	Overall
n		43
Group (%)	Healthy	21 (48.8)
	Tumor	22 (51.2)

**Table 4 biomolecules-13-00360-t004:** Patients back ground for TCGA.

	Level	Healthy	Tumor	*p*
n		8	290	
Age (mean (SD))		66.0 (12.4)	60.1 (13.5)	0.252
gender (%)	FEMALE	4 (50.0)	95 (32.8)	0.522
	MALE	4 (50.0)	195 (67.2)	
height (mean (SD))		166.6 (14.0)	167.8 (11.2)	0.812
weight (mean (SD))		66.3 (13.5)	72.9 (19.7)	0.382
race_list (%)	AMERICAN INDIAN OR ALASKA NATIVE	0 (0.0)	1 (0.3)	0.221
	ASIAN	1 (12.5)	117 (40.3)	
	BLACK OR AFRICAN AMERICAN	0 (0.0)	15 (5.2)	
	Not clear	1 (12.5)	7 (2.4)	
	WHITE	6 (75.0)	150 (51.7)	
vital_status (%)	Alive	1 (12.5)	222 (76.6)	<0.001
	Dead	7 (87.5)	68 (23.4)	
relative_family_cancer_history (%)	NO	3 (37.5)	157 (54.1)	0.15
	Not clear	0 (0.0)	41 (14.1)	
	YES	5 (62.5)	92 (31.7)	
neoplasm_histologic_grade (%)	G1	2 (25.0)	43 (15.1)	0.198
	G2	6 (75.0)	136 (47.7)	
	G3	0 (0.0)	96 (33.7)	
	G4	0 (0.0)	10 (3.5)	
child_pugh_classification_grade (%)	A	3 (37.5)	163 (56.2)	NaN
	B	2 (25.0)	15 (5.2)	
	C	0 (0.0)	0 (0.0)	
	Not clear	3 (37.5)	112 (38.6)	
fetoprotein_outcome_value (mean (SD))		7525.4 (8145.2)	15,969.3 (122,230.0)	0.845
platelet_result_count (mean (SD))		59,476.7 (158,012.4)	23,715.6 (67,091.8)	0.159
prothrombin_time_result_value (mean (SD))		6.9 (4.2)	4.0 (4.0)	0.04
albumin_result_specified_value (mean (SD))		8.2 (8.2)	26.0 (305.0)	0.869
creatinine_value_in_mg_dl (mean (SD))		1.2 (0.7)	3.0 (11.1)	0.652
TNM_T (%)	1	4 (50.0)	138 (47.6)	0.804
	2	1 (12.5)	72 (24.8)	
	3	2 (25.0)	67 (23.1)	
	4	1 (12.5)	10 (3.4)	
	Not clear	0 (0.0)	2 (0.7)	
	X	0 (0.0)	1 (0.3)	
TNM_N (%)	0	5 (62.5)	194 (66.9)	0.012
	1	1 (12.5)	2 (0.7)	
	Not clear	0 (0.0)	1 (0.3)	
	X	2 (25.0)	93 (32.1)	
TNM_M (%)	0	6 (75.0)	205 (70.7)	0.981
	1	0 (0.0)	4 (1.4)	
	Not clear	0 (0.0)	1 (0.3)	
	X	2 (25.0)	80 (27.6)	

## Data Availability

Not applicable.

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
