# Peer review of "HAMP as a Potential Diagnostic, PD-(L)1 Immunotherapy Sensitivity and Prognostic Biomarker in Hepatocellular Carcinoma"

_biomolecules, 2023, doi:10.3390/biom13020360_

Round 1

Reviewer 1 Report

Dear Editor,

 The authors showed HAMP can be a new candidate for biomarker of sensitivitiy of HCC after PD-L1 immunotherapy.. But some revisions are nessesary for publication.

Major points

1.      In this paper, there are no figures or schemes of patients back ground. Please provide figures or tables about patients.

2.     The prognosis of HCC is determined by liver function. Please provide Child-pugh scores of these patients.

3.     Recently, PD-L1 therapy is not effective for NASH related HCC. Please show cause of HCC of these patients.

Overall recommendation: 

 Major revision

Author Response

1. In this paper, there are no figures or schemes of patients back ground. Please provide figures or tables about patients.

Response: Thanks for your constructive suggestion. We have added tables about patients back ground in the “Supplementary Materials.docx which was uploaded as the attachment.

2. The prognosis of HCC is determined by liver function. Please provide Child-pugh scores of these patients.

Response: Thanks for your kind suggestion. For this, Child-pugh scores of these patients were showed in TCGA data from “Supplementary Materials.docx”.

3. Recently, PD-L1 therapy is not effective for NASH related HCC. Please show cause of HCC of these patients.
Response: Thank you very much for your kindly instructions.The causeof HCC is indeed a matter of concern in research. Unfortunately, GEO and TCGA databases do not provide us with such data. 

Reviewer 2 Report

Thanks for providing an opportunity to review the article, "HAMP is a Potential Diagnostic, PD-(L)1 Immunotherapy Sensitivity and Prognostic Biomarker in Hepatocellular Carcinoma" by Chen G et al., The authors should be commended for the work. However, I have a few suggestions/comments to make it a better piece.

Methods: Can authors outline more details on GEO database. The GEO should be expanded in full form prior to being abbreviated. Also, I am a little bit concerned about the small sample size the conclusions were drawn.

Discussion: I would recommend a major revision on the discussion section. The first 3 lines in the discussion section are to be deleted.

The sentence- "Whereas, in this research, we identified that HAMP also has a non-negligible role in the diagnosis of HCC based on analysis of variance, Lasso regression, and model construction" in the discussion section is very confusing and is contradicting the conclusions made by the authors. 

Author Response

  1. Methods: Can authors outline more details on GEO database. The GEO should be expanded in full form prior to being abbreviated. Also, I am a little bit concerned about the small sample size the conclusions were drawn.
    Response:Thank you very much for your instructions. We have added the full name of GEO which can be found in the revised manuscript. In addition, we would like to apologize for the small sample size and hope that more data can be available to supplement it in the future.
  2. Discussion: I would recommend a major revision on the discussion section. The first 3 lines in the discussion section are to be deleted.
    Response: Thank you very much for your kind reminder. After accepting your kindly suggestion, wehave deleted the first 3 lines in the manuscript.
  3. The sentence- "Whereas, in this research, we identified that HAMP also has a non-negligible role in the diagnosis of HCC based on analysis of variance, Lasso regression, and model construction" in the discussion section is very confusing and is contradicting the conclusions made by the authors.
    Response: Thanks for your constructive suggestion. After our careful observation, we have revised this issue in the discussion section.

Round 2

Reviewer 1 Report

Dear Dr. 

Editor, 

Overall recommendation: 

 Accept 

Final comments:

   The authors have revised point by point response and I think this paper is good for publication in this present form. 

Author Response

 Thank you very much for your comments.

Reviewer 2 Report

The authors have satisfactorily addressed the reviewer's comments.

Author Response

 Thank you very much for your comments.